# The First Whole Genome Sequence and Characterisation of Avian Nephritis Virus Genotype 3

**DOI:** 10.3390/v13020235

**Published:** 2021-02-03

**Authors:** Paula Lagan Tregaskis, Ryan Devaney, Victoria J. Smyth

**Affiliations:** Virology, Veterinary Science Division, Agri-Food and Biosciences Institute, Stormont, Belfast BT4 3SD, Northern Ireland, UK; ryan.devaney@afbini.gov.uk (R.D.); victoria.smyth@afbini.gov.uk (V.J.S.)

**Keywords:** avian nephritis virus, astrovirus, phylogenetics, molecular characterisation, genotyping and recombination

## Abstract

Avian nephritis virus (ANV) is classified in the *Avastroviridae* family with disease associations with nephritis, uneven flock growth and runting stunting syndrome (RSS) in chicken and turkey flocks, and other avian species. The whole genome of ANV genotype 3 (ANV-3) of 6959 nucleotides including the untranslated 5′ and 3′ regions and polyadenylated tail was detected in a metagenomic virome investigation of RSS-affected chicken broiler flocks. This report characterises the ANV-3 genome, identifying partially overlapping open reading frames (ORFs), ORF1a and ORF1b, and an opposing secondary pseudoknot prior to a ribosomal frameshift stemloop structure, with a separate ORF2, whilst observing conserved astrovirus motifs. Phylogenetic analysis of the *Avastroviridae* whole genome and ORF2 capsid polyprotein classified the first complete whole genome of ANV-3 within *Avastroviridae* genogroup 2.

## 1. Introduction

Astrovirus is considered an important human and animal pathogen known to cause gastroenteritis and other systemic diseases. The RNA virus has also been associated with neurological disease in mink, ovine and bovine species, and immunocompromised humans [1,2,3,4]. The *Astroviridae* family is split into two genera, namely *Avastrovirus* (of birds) and *Mamastrovirus* (of mammals). *Avastrovirus* disease was firstly recognised in 1965 as a hepatitis disease that occurred in ducklings, later described as an astrovirus-like agent in 1984 [5,6]. The *Avastrovirus* genus is composed of three genogroups designated *Avastrovirus* group 1 to 3 with avian nephritis virus (ANV) placed in *Avastrovirus* genogroup 2. Infection by ANV affects birds globally with pathogenesis ranging from subclinical infection to death, depending on the age of the bird and differing field strains [5,7,8,9,10,11,12,13,14]. Typically, ANV infection causes enteric and kidney disease in chicks and young birds; symptoms may include diarrhoea, uricosis (gout), stunting, tubulonephrosis, interstitial nephritis and mortality [11,15,16,17,18].

The isolation of ANV first occurred in 1979 from a chicken faecal sample, from a flock in which the infection was subclinical but caused pathological kidney changes in chicks [7]. Although originally described as an enterovirus-like picornavirus, reclassification after molecular characterisation identified ANV as a new member of *Astroviridae* [19], subsequently termed ANV-1. Under a negative stain electron microscope image, the astrovirus measures 28–30 nm and can be seen as a five-pointed star, although this conformation is reflective of pH [20,21]. A second ANV was identified in 1989, now known as ANV-2 as it is serologically distinct from ANV-1 [22,23]. 

Structurally, ANV is a small, round, non-enveloped virus containing a single-stranded positive sense RNA genome. The astrovirus genome, including that of ANV, is structured as three open reading frames (ORFs) situated between 5′ and 3′ untranslated regions (UTR) with a 3′ polyadenylated tail (poly-A tail) [24,25]. ORF1a is located after the short 5′ UTR, followed by overlapping ORF1b coding for two non-structural polyproteins (nsp1a and nsp1b) that are subsequently cleaved by proteolytic processing into the non-structural proteins necessary for genome replication [26]. The cleaved smaller peptides of ORF1a form a viral serine protease with a nuclear localisation bipartite signal motif (NLS) whereas ORF1b, through a frameshift translational mechanism, codes for the protein RNA-dependant RNA polymerase [24,27,28]. ORF2 encodes the structural capsid polyprotein protein that is initially cleaved by caspases, then trypsin enzymes. The N-terminal structural basic domain forms the capsid structure, whereas the hypervariable C-terminus contains the capsid spikes and acidic domains [26,29]. 

To date, three known serotypes of ANV exist, as designated from indirect immuno-florescence and virus neutralisation assays [11,14,23,30,31] and are numbered ANV-1, -2, and -3. Originally, ANV was documented as the only known avian astrovirus to be cultivated in primary chicken kidney cell culture, whereas other astroviruses were required to be adapted and trypsin activated to cell culture from isolates of embryonated eggs [8,28,32]. Contrary to this, further studies have documented difficulties in virus isolation of ANV, which in turn have prevented the necessary serological assays required for serotyping [11,31]. Classification has since relied on genotyping and inherent levels of ANV diversity suggest that the genotypes may be serologically distinct. The high level of diversity in positive sense RNA astroviruses occurs from the accumulation of genetic drift by point mutations and genetic shift through the mechanism of recombination, creating novel variants [33,34,35,36,37]. A study of ANV capsids circulating in UK poultry flocks reported that genetically diverse ANVs were found in the same flocks sampled at different times and ANV co-infections in the same flock were not uncommon [31]. Using group (genotype) clustering levels of 80% amino acid homology, eight potential groups (genotypes) have been identified [31,38] but, more recently, the eight genotypes were extended further to nine genotypes with an adapted criterion based on International Committee on Taxonomy of Viruses (ICTV) pairwise distance (p-distance) criteria [39].

Enteric disease in the poultry industry is responsible for huge economic losses worldwide, with a malabsorption disease, named runting–stunting syndrome (RSS), of partially determined aetiology, resulting in major culls of young flocks [30,40,41]. Commercial RSS-affected broiler flocks display poor feed conversion with poor growth rates, and are managed through culling of affected birds. Collectively, studies have detected numerous viruses which may contribute as co-infections to cause RSS, including ANV, chicken astrovirus (CAstV), fowl adenovirus (FAdV), chicken parvovirus (ChPV), infectious bronchitis virus (IBV), group A avian rotavirus (AvRT), and avian reovirus (ARV) [36,42,43,44,45,46,47,48,49]. Although ANV 1-3 genotypes, which appear by phylogenetic clustering based on amino acid homologies and approximate to ANV 1-3 serotypes, are detected in non-RSS-affected flocks, a recent study found that only the ANV-2 and ANV-3 genotypes were detected in samples from a small number of RSS-condemned flocks with ANV-2 viral detection at a higher abundance, indicating a possible greater pathogenetic effect [47]. 

Experimental infection studies of chicks concluded that age is an important factor in ANV pathogenicity, as the younger the is chick when infected, the more susceptible the chick is to developing disease [8]. A study of chicks experimentally inoculated with ANV at hatch reported clinical observations of stunting, enteric changes, diarrhoea, and discoloration of the kidneys [19]. The Narita et al. study (1990b) demonstrated similar results, although several chicks that died post-infection also had heavy urate deposits throughout their bodies, specifically on the peritoneum, liver, heart, and the leg tendons [17]. The histological findings were of renal degeneration and inflammation, lesions from the necrosis of tubular cells, and signs of pneumonitis around the parabronchi [17]. Other manifestations observed in the embryos included dwarfism, gelatinous consistency, oedema, haemorrhages, lesions, swollen kidneys, dilated ureters, urate deposits across the viscera, heart, and kidneys, and mortality [14,50,51,52].

Transmission of ANV is predominantly horizontal although it is also tentatively assumed to be vertically transmitted. For instance, a research study detected ANV in dead-in-shell duck embryos in the absence of other known duck viral diseases [53]. Interestingly, an experimental challenge in embryos from parents vaccinated with the ANV-2 serotype (M-8 strain also classified as genotype ANV-2) and non-vaccinated breeding hens indicated that ANV maternal antibody protection occurred, preventing pathological damage and the replication of virus in the embryos’ kidneys but not the intestines [23,50]. This experimental vaccine resulted in maternal antibody levels being elevated, allowing good ANV protection, however, the antibody protection from natural ANV infections was observed to give a lower level of protection but reduced mortality [50,54]. As alluded to by Todd et al., a vaccine would need to encompass all the antigenetically diverse ANV strains to be commercially viable [31]. 

This study presents the first molecular characterisation of the complete ANV-3 genome sequence isolated from an RSS-affected UK broiler flock. The genetic characterisation demonstrates a partially overlapping ORF1a and ORF1b and a non-overlapping ORF2, conserved domains, and a heptameric ribosomal frameshift (RFS) mechanism. A review of the phylogenetic classification study of the ANV capsid amino acid ORF2 region and whole genome of *Avastrovirus* has accentuated the difficulties in the assignment of ANV genotypes.

## 2. Materials and Methods

### 2.1. Sample Preparation and Next Generation Sequencing 

The gut contents of 2 to 7 birds with RSS from each of 5 flocks of broiler chickens aged 13–31 days and from 2 normal flocks were pooled per flock and prepared, followed by viral enrichment, as described in Devaney et al. (2016) [47]. In order to provide sufficient amounts of nucleic acids for the production of next generation sequencing (NGS) libraries, nucleic acids from each pooled flock sample underwent both whole genome amplification (WGA) and whole transcriptome amplification (WTA) using REPLI-g kits (Qiagen, Hilden, Germany). The amplified nucleic acids (DNA and complementary DNA (cDNA) from the 7 flock samples were used to make 7 DNA and 7 cDNA libraries. Briefly, the 14 libraries were sequenced by de novo methods on a MiSeq instrument (Illumina, San Diego, USA) using the Nextera XT library preparation kit v3 (600 cycles) (Illumina) and run as 300 nt paired end reads as set out by the manufacturer’s kit instructions. MiSeq sequencing reads were assembled in Basespace (Illumina) using the velvet assembly tool v1.0.0 [55,56]. Assembled contigs (contiguous sequences) were identified through comparison to existing sequences using the NCBI Basic Local Alignment Tool (BLAST) non-redundant database (nr) [57], visualised using Metagenome Analyser v5.7.1 (MEGAN) [58,59], and uploaded to the metagenomics analysis server MG-Rast [60] http://metagenomics.anl.gov/linkin.cgi?project=17287, accession number 4689855.3 (VF14-92 A2), as described by Devaney et al. (2016) [47]. The ANV genome was further analysed in Geneious v10 (Biomatters Ltd., Auckland, New Zealand) and identified gaps were filled by conventional Sanger sequencing (patch PCR). The NGS contigs and patch PCR of ANV-3 full genome sample VF 14-92-A2 presented a low depth of coverage. To confidently confirm the ANV-3 whole genome sequence (WGS), a positive ANV-3 sample with a higher abundance of reads, from a subsequent NGS study, was also characterised. The VF16-03-164b sample came from a poorly performing broiler flock (day 11 chicks), upon which WGS was performed to gain a greater genome coverage depth. The WGS results of both ANV-3 strains were submitted to NCBI with accession numbers MT585643 (VF14-92/A2) and MT585644 (VF16-03/164b). 

### 2.2. Sequencing, Phylogenetic, and Recombination Analysis

The alignment for primer design was carried out in Geneious v10 (Biomatters Ltd.). Sanger sequencing was performed commercially. The assembled contigs were mapped to reference strains of ANV-1 from China (HM029238) and pigeon ANV (HQ889774) to create a scaffold reference mapping genome. From the results of the NGS, it was observed that an almost complete genome of ANV was present from sample VF14-92-A2, which originated from a flock with RSS. Primers were designed to patch sequence the gaps in the partial genome and to identify the sequences at the UTR 5′ and 3′ ends of the ANV genome through Sanger sequencing (Table 1). The p-distance, multiple sequence alignment (MSA), and phylogenetic trees were completed in Mega X (version 10.0.4) [61] to avail of the alignment tool, Multiple Sequence Comparison by Log-Expectation (MUSCLE) and phylogenetic tree builder through maximum likelihood (ML) with 1000 bootstrapping replicates used to generate phylogenetic trees with 145 Genbank *Avastrovirus* sequences aligned for the capsid and 78 sequences for WGS analysis. Phylogenetic trees were graphically viewed and edited in Interactive Tree of Life [62]. The protein motifs were assigned through Introproscan [63] and annotated in Geneious (Appendix A and Figure 1a,b). To analyse the highly variable ANV ORF2 capsid for the detection of recombination signals, recombination detection program 4 v4.100 (inclusive of recombination detection program (RDP) methods; RDP, GENECONV, BootScan, MaxChi, SiScan, LARD, and 3SEQ) was utilised, where accepted recombination events had *p*-values of ≤0.05 from over four methods after Bonferroni correction [64,65,66,67,68,69,70,71,72]. 

### 2.3. ANV-3 Sequence Confirmation 

To confirm the ANV-3 genome width coverage, whilst achieving an appropriate read depth coverage, a sample with a higher read abundance of ANV-3 determined by DIAMOND [74] and MEGAN analysis was selected. Initially, this sample, VF16-03-164b, was processed by Trim Galore [75] to a Q ≥30 score with validated reads viewed on MultiQc [76]. To map the validated VF16-03-164b reads, the read aligner program BWA-MEM [77] with VF14-92-A2 as the reference sequence was utilised through Galaxy GVL [78] via CLIMB [79]. The Tablet program [80] enabled the visualisation of the ANV-3 reference sequence VF14-92-A2 and aligned it to VF16-03-164b for genome coverage and read depth. As a secondary confirmation of depth and genome width coverage, the reads of VF16-03-164b were also mapped to VF14-92-A2 ANV-3 WGS in Geneious by 5 iterations in high sensitivity mode (Figure 4a,b). 

## 3. Results

The ANV-3 VF14-92-A2 strain genome, at 6959 nt in length, is characterised by an overlapping genetic region of ORF1a (3045 nt) and ORF1b (1530 nt) wherein lies the conserved ribosomal frameshifting mechanism, with ORF2 (2040 nt) reading in the same frame as ORF1a, and all of which are positioned within a short 5′ UTR and a longer 3′ UTR (319 nt) inclusive of the 3′poly-A tail (Figure 1a). The ORF1a polyprotein M_r_ is calculated at 113.451 kilo Daltons (kDa), ORF1b polyprotein is 59.084 kDa, and ORF2 polyprotein is 73.872 kDa. The 5′ UTR is 15 nt long and begins with a characteristic sequence conserved across Avastrovirus species, CCGAA, followed by ORF1a reading in frame 1, starting at position 16 nt to 3060 nt encoding a polyprotein of 1014 amino acids (aa). A conserved peptidase domain, the homologous Superfamily 1 trypsin 2-like peptidase domain, was identified mid-ORF1a from 517 to 642aa (pfam13365).

A conserved bipartite nuclear localisation signal (NLS) motif comprising KKKGKTK and TEEEY motifs (761 to 765aa) is also present in ORF 1a (Figure 1a). The conserved replication feature of astrovirus is a frameshift structure mechanism where an overlapping ORF1a and ORF1b region occurs due to a shared heptameric sequence known as a “slippery sequence” which creates a –1 ribosomal frameshift signal (RFS) (Figure 1b) where the ribosome shifts, in the 5′ direction, by one nucleotide [81]. This frameshift translation mechanism requires two cis-acting sequence elements, the heptameric slippery sequence and the stemloop secondary structure [27,82]. In strain VF14-92-A2, the ORF1a stop codon is located at nt positions 3058-3060 immediately downstream of the ORF1b partial overlapping polyprotein heptameric slippery sequence which begins at nt position 3051, ending at 3057 (AAAAAAC), which is followed by a stemloop structure sequence between position 3066–3079 nt (the underlined nucleotides form the base-pairing structure of the stemloop: CCCCCUUCGGGGGGC). 

The translation continues after the –1 base ribosomal frameshift, from frame 1, to continuing translation from the lysine aa by shifting to reading in frame 3 (Figure 1b). Notably, no AUG start codon (methionine) exists immediately preceding the heptameric slippery sequence in ANV, as seen in chicken astrovirus, however, a start codon is present in both ANV and chicken astrovirus located from 12–17 nt upstream of the slippery sequence, although is only translatable into methionine in ANV in frame 2, which is non-viable due to the presence of multiple stop codons downstream (Figure 1b). A start codon is also found before the stemloop in chicken astrovirus and duck astrovirus, although this is intermittent in ANV, and whilst present in strain VF16-03-164b, it is absent from VF14-92-A2 (Figure 2). The goose astrovirus genotypes differed in comparison to the other Avastrovirus species, with no start codons before the heptameric slippery sequence or stemloop regions. 

The VF14-92-A2 stemloop motif was confirmed in the RNAfold program with minimum free energy (MFE) modelling (http://rna.tbi.univie.ac.at) (Figure 3a) with a pseudoknot observed prior to the slippery sequence, creating a more complex configuration to increase the efficiency of the RFS by slowing or stalling the ribosome and was confirmed in the ANV-3 VF16-03-164b sequence (Figure 3b).

The VF 14-92-A2 isolate possesses a short overlap of only 10 nt, in which the heptameric RFS is located, at the 3′ end of ORF1a and 5′ end of ORF 1b, shifting the reading frame of ORF 1b, which codes for the polyprotein RNA-dependant RNA polymerase (RdRp) of 509aa (Figure 1a,b). As seen in other astroviruses, the RdRp centre forms the conserved Supergroup 1 motifs [19,27,83]. The RdRp is translated –1 in frame 3 and ANV-3 RdRp motifs were mapped at DWTRFD (3828 to 3845 nt), GNPSG (4011 to 4025 nt), YGDD (4161 to 4172 nt), and FGMWVK (4245 to 4262 nt) and confirmed by Interpro PFam analysis (Clan RdRp CL0027 and CD cd01699 RNA dep RNAP) (Appendix A). The conserved avastrovirus spacer gap (Figure 1a), although not present in passerine Avastrovirus, is a short 19 nt stretch between ORF1b and the ANV-3 capsid ORF2 and is present in both ANV strains. It facilitates a second frameshift, allowing ORF 2, which codes for a 680aa polyprotein, to switch to frame 1 during translation (4600 to 6639 nt), whereas, in other mammalian and reptile astroviruses, ORF1b and ORF2 overlap [84,85]. The relatively long 3′ UTR of 303 nt, excluding the polyadenylated tail, with a complex stemloop II motif, when analysed by RFam, matches a similar coronavirus 3’ stemloop II-like motif (s2m), as seen in other astrovirus species and ANV strains, although not detected in turkey astrovirus genogroup 2 [86]. The s2m is located from 78 to 120 nt in the UTR (nt position 6716 to 6756) and the RNA secondary structure was confirmed through MFE modelling (Appendix A) [87].

### 3.1. Confirmatory NGS Genome Coverage

The VF14-92-A2 WGS, as a reference strain, mapped reads from a confirmatory sample (VF16-03-164b) to yield full genome width and depth coverage, although, as anticipated, the coverage was uneven and the 5′ and 3′ UTRs returned low coverage (Figure 4a,b). The Geneious reference mapping algorithm, from 5 iterations, mapped 9241 reads. The ORF1a 3′ region had the highest coverage at a 1042 read depth. The BWA-MEM algorithm and Tablet visualisation showed a similar profile to confirm the presence of the ANV-3 genome.

### 3.2. Amino Acid Percentage Homology, Pairwise Distance Percentage Homology, and Recombination 

The complete VF14-92-A2 ANV-3 capsid polyprotein 680 aa sequence aligned with other Avastrovirus sequences deposited in Genbank (NCBI) and designated the VF14-92-A2 and VF16-03-164b strains in the same clade with 92.4% aa homology and as an ANV-3 genotype (Figure 6). The comparison of ANV-3 VF14-92-A2 as an intra-genotype aa percentage identity resulted in a homology similarity of 83.3% to KM985691 and 94.9% to HQ330481. Within the capsid polyprotein alignment, the most divergent ANV isolate to VF14-92-A2 found ANV genotype 6 KU711057 Brazil 19-5 at a low 54.6% nucleotide homology.

The Avastrovirus genogroup classification by p-distance criteria is currently under review by the ICTV, however, it sets the demarcation criteria of the p-distance limit between genogroups as 0.704 + 0.013 https://talk.ictvonline.org [84]. Currently, the ICTV classifies all ANV strains by aa p-distance of the capsid Avastrovirus as intra-genogroup 2, including the sequences from this study, ranging from 0 to 0.588 and with an inter-genogroup p-distance range of >0.7290 to 1.45 (Appendix A). The aa p-distance of VF14-92-A2 ANV-3 to the most divergent isolate of Avastrovirus genogroup 2 ANV was the Brazilian strain, ANV genotype 6, KU711057, at 0.551. Within the ICTV Avastrovirus genogroup 2, the closest homology to ANV-3 VF14-92-A2 was from ANV UK strains HQ330481 and HQ330494, with a p-distance of 0.041 (Figure 5 and Figure 6 and Appendix A). The Avastrovirus whole genome nucleotide MSA p-distance was calculated to reveal that within the ANV genogroup, the p-distance ranged from 0 to 0.309, and between other Avastrovirus inter-group clades from 0.340 to 1.019 (Appendix A).

The genetic recombination analysis of ANV capsid strains revealed putative recombination signals as common events, with some strains even possessing multiple recombination events. Specifically, the RDP4 program found that the ANV-3 strains VF14-92-A2 and VF16-02-163b both had a putative recombination signal at the C-terminus region of ORF2 (Appendix A). An acceptable p-value of less than 0.05 was generated from RDP4 programs RDP, GENECONV, BootScan, MaxChi, SiScan, Phylpro, LARD, and 3Seq, with VF14-92-A2 *p*-values of 9.58 × 10^−3^ 9.005 × 10^−1^, 3.854 × 10^−2^, 2.053 × 10^−5^, 1.001 × 10^−16^, 3.275 × 10^−4^, 1.327 × 10^−6^, and 3.275 × 10^−4^, and VF16-03-164b p-values of 1.031 × 10^−10^, 1.830 × 10^−5^, 1.881 × 10^−9^, 1.558 × 10^−10^, 5.135 × 10^−19^, 3.885 × 10^−20^, 5.810 × 10^−31^, and 3.885 × 10^−20^, respectively. The breakpoints in each strain were both similar, with the VF14-92-A2 nucleotide position region occurring at 1374–2258 and for VF16-03-164b at positions 1298–2252, however, the identified major and minor parental donors were from different intergenic clades. The VF14-92-A2 potential major parent was KU711052, an ANV-8 strain, and the potential minor parent KM985698 VIC-3c, an ANV-1 strain. However, the VF16-02-164b inferred unknown potential parent was HO086767, an ANV-1 strain, and the minor parent was KM985694 NSW-4a, also an ANV-1 strain. The parental donors speculated in this study were the most closely related recombinant donors available from the multiple sequence alignment. Further genomic screening and sequencing of circulating ANV strains may help to provide epidemiological relevance to the recombination signals found in this study.

## 4. Discussion

The ANV-3 whole genome sequence characterisation is the first report of the ANV genotype 3 virus discovered, through a dual approach of NGS and Sanger sequencing methods, from an RSS-affected flock. This study sought to characterise the ANV-3 isolate VF14-92-A2 full length genomic sequence which encompasses 6959 nt, inclusive of the poly-A tail (MT585643), with an additional confirmatory whole genome sequence discovered in samples from a poorly performing broiler flock VF16-03-164b (MT585644). 

The genomic organisation of ANV-3 VF14-92-A2 is typical of other ANVs, commencing with a short 5′ UTR of 15nt and ending with a much longer 3′ UTR of 319 nt, which is also a typical feature of Avastrovirus, although shorter than the typically longer UTR regions in the Mamastrovirus genus [86], and followed by a polyadenylated tail. Between these extremities are situated three ORFs, the first and third of which, ORF 1a and ORF 2, are in the same frame, while the intervening ORF, ORF 1b, is in a different frame, which is facilitated by the overlapping region of 10 nt at the 3′ end of ORF 1a and the 5′ start of ORF 1b, that contains the conserved heptameric slippery sequence necessary for translational frameshifts (Figure 2) and is present in all ANVs. Between ORF 1b and ORF 2, there is the conserved, amongst most avastrovirus species, short 19 nt spacer region (Figure 1a,b), which affects a frameshift when the positive sense single-stranded RNA genome is translated directly as messenger (m)RNA. During infection, the replication polyproteins from these three ORFs are proteolytically cleaved to form the RdRp, a protease, and capsid proteins [25,86]. 

The NLS bipartite region identified in ANV-3 VF14-92-A2 within ORF1a aa consists of conserved motifs KKKGKTK and TEEEY (Figure 1a and Appendix A), with similar NLS representations in other ANV-1 and ANV-3 strains and in chicken astrovirus sequences. However, historically, the bipartite motifs were proposed to have a spacer gap of only around 10 aa, whereas those of the ANV and CAstV are separated by a 23–24 aa gap [85,87,88,89]. A second conserved region of ORF1a is similar to a region within the serine endopeptidase S32 of equine arteritis virus, of the order *Nidovirales*, with serine peptidase DNA-binding activity and formation of protein dimers (Figure 1a and Appendix A) [90].

An RFS mechanism occurs to genetically express different proteins coded by overlapping ORF mRNA during translation by changing the reading frame, shifting by one base in the 5′(−1) or 3′ (+1) direction [91]. The RFS is observed as the translational shift “slippery site” with a short spacer sequence and a downstream secondary structure, usually either a stemloop or pseudoknot. The astrovirus ORF1a 3′ end position is a partially overlapping ORF1a/1b junction with a conserved “slippery heptameric” sequence, 5′-AAAAAAC-3′, and upstream of a stemloop motif to create the RFS (Figure 1b, Figure 2 and Figure 3). The overlapping frameshift is well documented in astrovirus, however, this analysis of the ANV-3 strains found a potential pseudoknot prior to the slippery sequence and stemloop of ORF1b, which creates a more stable and efficient RFS translating from the overlap of the reading frames at the AAA codon, lysine, by switching –1 base/frame coding the polyprotein RdRp (Figure 1b, Figure 2 and Figure 3). The AUG codon at the start of ORF1b, prior to the slippery sequence, is typically observed in mammalian astroviruses and CAstV; however, it is not always conserved in Avastrovirus, and is absent from all ANV genotypes, although an AUG codon is situated incidentally 12 nt upstream of the slippery heptameric sequence in both ANV and CAstV, and earlier than the beginning of the ORF1b lysine translation, which may provide a methionine towards the C-terminal end of the ORF1a polyprotein. Furthermore, preceding the stemloop sequence, a putative methionine is found intermittently in ANV strains, where the ANV-3 VF14-92-A2 genotype strain has an aa deletion gap, and a methionine is present in the VF16-03-164b strain and all the DAstV strains, whilst, in contrast, the GAstV methionine is situated after the stemloop (Figure 2). The significance of these methionine sites as putative ribosomal initiators in the RFS mechanism remains to be determined. 

The RdRp is conserved in all RNA viruses without a DNA stage in replication in order to synthesise a complementary RNA strand from the RNA template [19,27]. The RdRp enzyme catalyses the replication of the RNA template to RNA and has an active catalytic centre formed of conserved motifs known as Koonin’s Supergroup I DWTRFD, GNPSG, YGDD, and FGMWVK (Figure 1a and Appendix A) [19,27,83], which are present in both of the ANV strains described herein. The astrovirus capsid gene, ORF2, normally overlaps with ORF1b, however, the ANV-3 strains and the majority of Avastrovirus strains possess a spacer gap whilst switching frames. In the ANV strains VF14-92-A2 and VF16-03-164b from this study, the spacer gap was between 13 to 19bp. 

The highly conserved 3′ UTR s2m motif is observed in positive sense RNA viruses with poly-A tails, such as the *Caliciviridae, Coronaviridae, Picornaviridae*, and *Astroviridae* [85,92,93]. The s2m may be transferred to different RNA families through non-homologous recombination events and, as RNA viruses have a high mutation rate, its conservation implies the s2m has an important function [93]. Comparison of the s2m structure to the microRNAs involved in RNAi-associated gene regulation hypothesises that the s2m function may be implicated in gene silencing [94,95]. The 3′ UTR was previously reported to contain a complex s2m in astroviruses and is similar to the coronavirus 3′ s2m structure [92] and this is also confirmed for the ANV-3 VF14-92-A2 and VF16-03-164b strains (Appendix A). 

The genomic region with the highest variability is the ANV capsid gene (ORF 2) sequence where homology classification of the ANV was incepted in 2010 to genetically group ANV-1 to -6 [96]. The ANV genotypes are categorised by the application of the capsid protein with >80% amino acid similarity as the cut-off per group. A later study extended this system to include genotypes 7 and 8 [38]. This estimate was based on the observed levels of similarity that existed between small numbers of representative strains in each of five of the proposed genogroups compared to a lower level of ORF2 amino acid homology of 71% shared between the ANV-1 and ANV-2 genotypes, which had been previously determined as serologically distinct [8,19,97]. As proposed by Todd, no correlation can be made when comparing ANV genotypes and the geographical location of the ANV host [31]. The Todd classification of >80% amino acid pairwise homology definition of a genotype (group) is problematic when considering the diverse Brazilian ANV strains [39], as many of these strains intra-genotype, such as KU711057, but present a low 71% homology within the ANV genotype 6 clade (Figure 5 and Figure 6). Given that 71% is also the amount of ORF2 shared amino acid homology between the two serologically distinct strains, ANV-1 and ANV-2, it may be that, in the future, the more variable C-terminal half of ORF2, which largely determines ANV antigenicity, assumes greater responsibility for ANV strain classification as opposed to the entire ORF2. 

The genetic diversity of RNA viruses is fundamentally driven by the error-prone RdRp point mutations at high error rates and also recombination events [98,99]. In this study, it was observed that the ANV capsids contained frequent putative recombination signals, where multiple ORF2 recombination events may have occurred within a single ANV strain. Both VF14-92-A2 and VF16-03-164b strains’ recombination signals displayed similar breakpoints in the C-terminus spike region which had previously been identified as a recombination hotspot [31]. Furthermore, the two ANV-3 strains’ potential parents were putative intergenic recombination events: VF14-92-A2 ANV-3 major parent ANV-8, and minor parent ANV-1, whilst VF16-03-164b intergenically recombined with ANV-1, suggesting that ANV-3 had possibly evolved from ANV-1. The antigenic shift, due to intergenic recombination within the capsid gene, and specifically within the spike coding region, may have increased the viral pathogenicity and viral antigenic variability, leading to further difficulties in phylogenetically mapping highly divergent ANV strains. 

Currently, the ICTV propose that speciation criteria of the *Avastrovirus* family are based on the capsid polyprotein aa mean p-distance between genogroups of 0.704+, https://talk.ictvonline.org [84]. The p-distance of the ANV-3 VF14-92-A2 capsid aa sequence falls within the *Avastrovirus* genogroup 2, which encompasses all ANV genotypes, although the classification of genotypes has not been specifically defined (Appendix A). The challenging endeavours to classify the highly recombinant and divergent astroviruses, combined with an expanding array of newly discovered novel sequences, has generated various classification interpretations. Previously, classification of genotypes by the p-distance of ANV in ICTV *Avastrovirus* genogroup 2 was overly stringent, with the criteria set at the genotype range for highly divergent RNA viruses (p-distance up to 0.284) which was too narrow. However, a recent study reclassified the ANVs into genotypes based on ICTV criteria in which they altered the p-distance genogroup range of 0.414 to 0.748 and expanded within the genotype to 0.047–0.299 [39]. This study found 11 ANV genotypes, as set out in distinct clades (Figure 5 and Figure 6), although the p-distance criteria by ICTV or Espinoza et al. did not succeed in separating out genotypes based on the ANVs’ ORF2 amino acid sequences [39,84]. Based on the *Avastrovirus* ANV capsid aa, applying the inter-genotype criteria created the occurrence of p-distance conflicts, for instance, HQ330486 ANV-2 placed in genotype 2 by the p-distance conflicts with a genotype 1, HQ330496, causing a low inter-genotype p-distance of 0.1894, with similar occurrences seen across most ANV genotypes (Appendix A). The *Avastroviridae* WGS analysis of the nt (but not aa due to translation occurring over differing frames) p-distance was unsuccessful in providing any resolution and designating appropriate genotype criteria (Figure 5 and Appendix A). 

The classification of the highly divergent ANV genotypes is challenging when adopting Todd’s, ICTV’s, or Espinoza’s classification suggestions, as inherent complications occur. Ideally, genetic phylogeny should reflect serological diversity in order to provide useful insight into pathogenicity and immunological protection. This study, by means to classify the ANV genotypes, has separated the ANV genotypes by the interpretations of distinct clades. The whole genome sequencing of an ANV-3 strain from RSS-affected broiler chickens has permitted the genetic characterisation of the virus and genome organisational structure by establishing the conserved domains found in *Avastrovirus* within this strain. Along with the other full length ANV strains, genetic characterisation will enhance taxonomic classification, aid molecular assays for diagnostic detection to be developed, and enhance further understanding of the epidemiology and pathogenicity of ANV. 

## Figures and Tables

**Figure 1 viruses-13-00235-f001:**
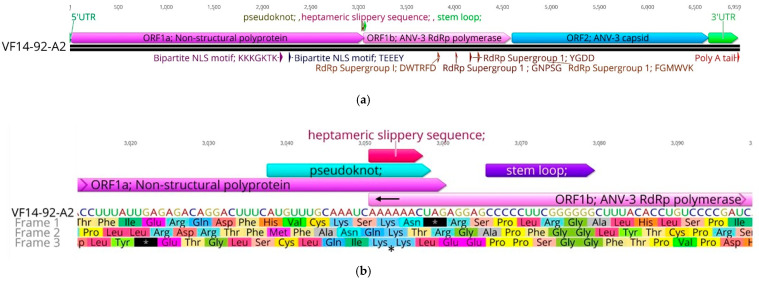
Schematic representation of ANV genotype 3 whole genome and ribosomal shift mechanism. (**a**) Schematic representation of ANV genotype 3 whole genome structural organisation comprising a partially overlapping open reading frame 1a (ORF1(a) and ORF1b and separate ORF2. Motifs and secondary structures located in ORF1a nuclear localisation signal NLS) bipartite KKKGKTK and TEEEY, ORF1b RdRp motifs DWTRFD, GNPSG, YGDD, and FGMWVK. (**b**) Schematic representation of the partially overlapping and –1 (signified by an arrow) ORF1a and 1b ribosomal frameshift signal with mapped pseudoknot, heptameric slippery sequence, and stemloop. The lysine amino acid (K) in frame 3 where translation recommences after the ribosomal frameshifting signal (RFS) is marked as *.

**Figure 2 viruses-13-00235-f002:**
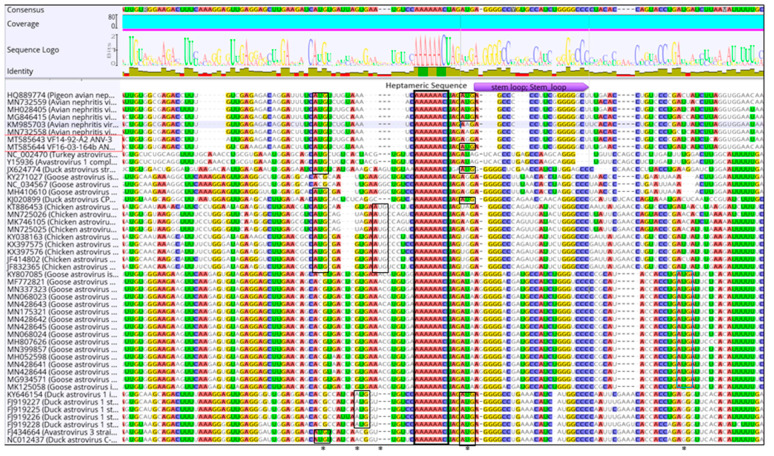
Astroviridae ribosomal frameshift mechanism ORF1a and ORF1b. Alignment of the Avastroviridae whole genome heptameric slippery sequence and stemloop region including ANV-3 strains and MT585643 VF14-92-A2 and MT585644 VF16-03-164b. Red box is ANV VF14-92-A2 and VF16-03-64b, blue box and * identify the methionine/putative start codons, and the black box marks the heptameric slippery sequence.

**Figure 3 viruses-13-00235-f003:**
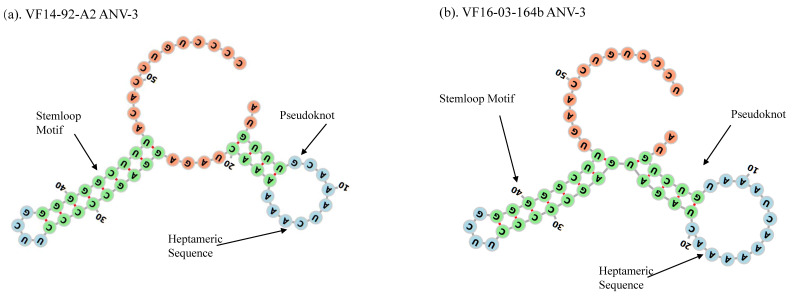
(**a**,**b**) RNAfold secondary structures of pseudoknot, heptameric slippery sequence, and stemloop with a minimum free energy (MFE) of -16.8 kcal/mol for MT585643 VF14-92-A2 and 164b -17 kcal/mol MFE for MT585644 VF16-03-164b.

**Figure 4 viruses-13-00235-f004:**
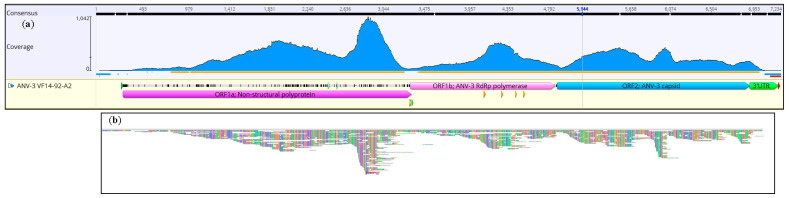
Genome coverage of ANV-3. (**a**) The whole genome coverage validation of ANV-3 reads from MT585643 VF14-92-A2 with MT585644 VF16-03-164b used as a reference sequence. The Geneious read mapping algorithm, set at 5 iterations, resulted in whole genome coverage with 9241 reads. The ORF1a 3′ region had the highest coverage at a 1042 read depth. (**b**) The BWA-MEM read aligner/Tablet visualisation showed a similar profile to Geneious and whole genome breadth and depth of coverage.

**Figure 5 viruses-13-00235-f005:**
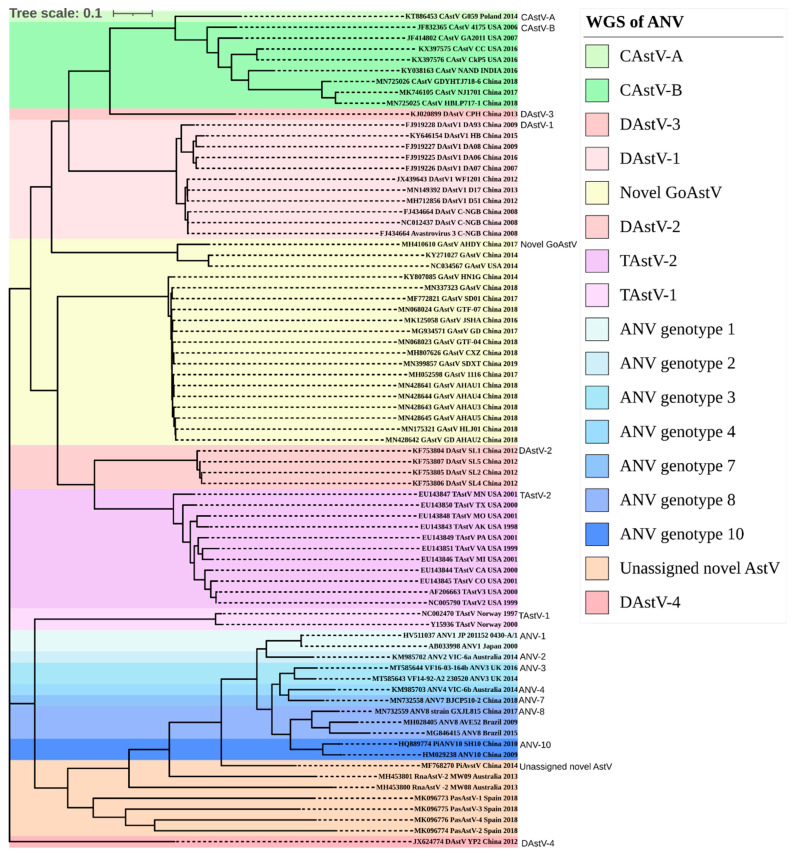
Astroviridae whole genome sequence (WGS) phylogeny. The Avastroviridae whole genome nucleotides of 78 sequences, starting from ORF1a phylogenetic analysis in MEGA X from MUSCLE-aligned sequences. Evolutionary history was inferred by maximum likelihood with bootstrapping of 1000 replicates. The tree was annotated into clades and ANV genotypes coloured blue with sequences from this study, with MT585643, VF14-92-A2, and MT585644 VF16-03-164b positioned in Avastrovirus genotype 3. Clade abbreviations; CAstV = Chicken astrovirus, DAstV = Duck astrovirus, GAstV = Goose astrovirus, ANV = Avian nephritis virus, and TAstV = turkey astrovirus.

**Figure 6 viruses-13-00235-f006:**
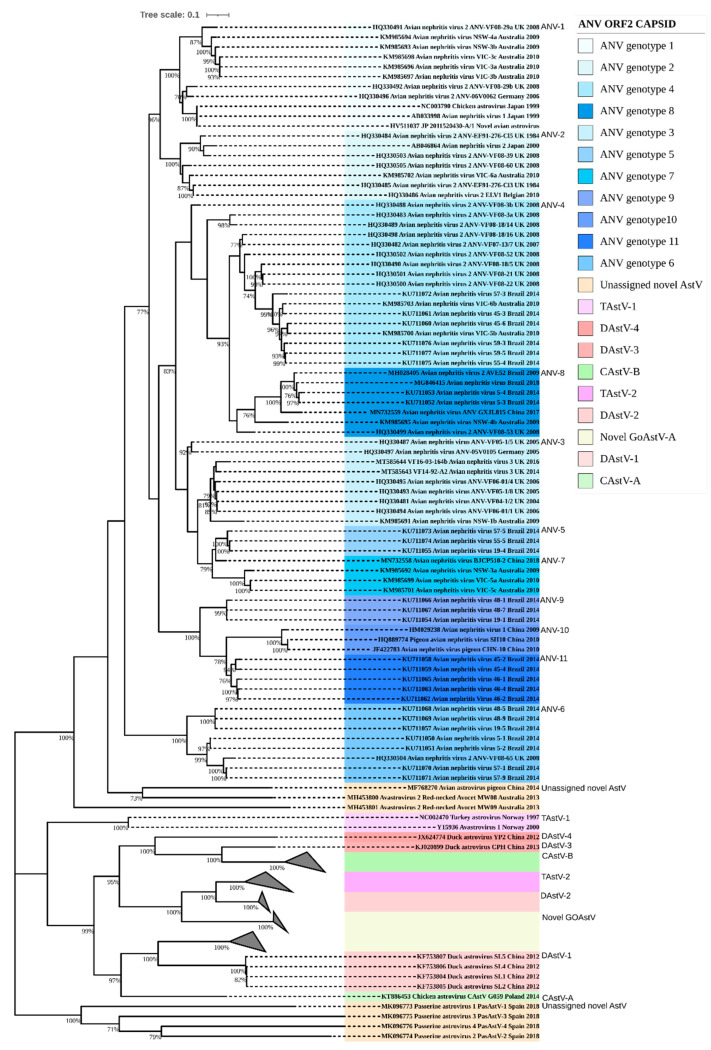
Astroviridae capsid ORF2 phylogeny. The Avastroviridae capsid ORF2 amino acids of 145 sequences phylogenetic analysis in MEGA X from MUSCLE-aligned sequences. Evolutionary history was inferred by maximum likelihood with bootstrapping of 1000 replicates. The tree is annotated in clades and ANV genotypes coloured blue with sequences from this study (in bold), with MT585643, VF14-92-A2, and MT585644 VF16-03-164b positioned in avian nephritis genotype 3. Clade abbreviations; CAstV = Chicken astrovirus, DAstV = Duck astrovirus, GAstV = Goose astrovirus, ANV = Avian nephritis virus, and TAstV = turkey astrovirus.

**Table 1 viruses-13-00235-t001:** Avian nephritis virus (ANV) primer sequence specifications. Sanger sequencing primers for patch sequencing of MT585643 VF14-92-A2.

Primer Name	Primer Sequence (5′-3′)	Annealing Temp (°C)	Amplicon Size in Base Pairs (nt)	Position in the ANV-3 VF14-92-A2 Genome (5′-3′) (nt)	Assay
1AF	GCGCAAAGTGACTCTAG	55	425	341–357	In-house
1AR	CAAGTGCCAGAGCTTC			749–764	
2AF	GCTTGGATTGACTACCAG	55	323	2363–2380	In-house
2AR	CTGAGCGCTGCTTCT			2671–2685	
3AF	AGGGATTGAACTTCCTG	54	332	4527–4543	In-house
3AR	TATCTGCCTAGTGAGACC			4841–4858	
4AF	CCTGAAGCTGTGTCCTA	51	365	4465–4481	In-house
4AR	GTCCAGAAATCGTACCAAG			4811–4829	
5AF	AGGTCATTTCCACCTACTC	54	777	5716–5816	In-house
5AR	TTCGAGTTGATCCACAC			6558–6574	
6AF	GCCCGGAGAAGGCGACT	53	345	6229–6245	In-house
6AR	TCGAGTTGATCCACACAATCAAACCTC			6547–6573	
5′ F	CCGAATAGATGGGATGGCT	50	321	1–19	In-house
5′ R	GTCATCACAGCCTTTTCCTC			302–321	
3′ F	GTAAACCACTGGYTGGCTGACT	50	271	6682–6703	[73]
3′R	TTTTTTTTTTTTAAAAGTTAGC			6931–6952	In-house
Pre-ORF2	ACCTTGAATCCCTGTGGGGCA	57	2535	4406–4426	[31]
Post-ORF2	AAAAGTTAGCCAATTCAAAATTAATTC			6914–6940	

## Data Availability

The datasets generated for VF14-92/A2 are available at MG-Rast http://metagenomics.anl.gov/linkin.cgi?project=17287, accession number 4689855.3 and NCBI Genbank with accession numbers MT585643 (VF14-92/A2) and MT585644 (VF16-03/164b).

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
