# Peer review of "The First Whole Genome Sequence and Characterisation of Avian Nephritis Virus Genotype 3"

_viruses, 2021, doi:10.3390/v13020235_

Round 1
Reviewer 1 Report
Good work in one of the understudied viruses with poorly understood role in disease pathogenesis. However, there are some gaps/questions that I have about this manuscript.
Title claims this the first whole genome sequence and characterization, yet in line 159 and figure 5 the authors used 78 sequences available on Genbank, please explain.
It appears that a large section in the results is missing, and that is: what is the results of the metagenomics whole genome sequencing from the 7 farms? Was ANV reads similarly prevalent all of them? While the manuscript is well written, it is not clear to me why ANV was picked as the relevant pathogen in all these RSS cases.
Lines 275 – 297: Do these similarities and differences with other available sequences make epidemiologic sense? And if they don’t, what is your explanation?
Lines 298 – 313: Do these hypothetical donor strains make epidemiological sense? And if they don’t, what is your explanation?
Author Response
Reviewer 1
Comments and Suggestions for Authors
Good work in one of the understudied viruses with poorly understood role in disease pathogenesis. However, there are some gaps/questions that I have about this manuscript.
Dear reviewer,
I would like to thank you for taking the time to review this study and for your comments.
Title claims this the first whole genome sequence and characterization, yet in line 159 and figure 5 the authors used 78 sequences available on Genbank, please explain.
The 78 whole genome sequences are sequences that fall within the Astroviridae including duck and turkey astrovirus, and avian nephritis virus genotypes. Only several ANV whole genomes were available on NCBI database and importantly no ANV-3 sequences were available until our study sequenced and submitted two ANV-3 whole genomes and is presented in the phylogenic tree Fig 5.
It appears that a large section in the results is missing, and that is: what is the results of the metagenomics whole genome sequencing from the 7 farms? Was ANV reads similarly prevalent all of them? While the manuscript is well written, it is not clear to me why ANV was picked as the relevant pathogen in all these RSS cases.
This study follows on from the published metagenomic study that involved RSS cases and is referred to by the reference number [47] and Devaney et al 2016 in the methods paragraph section 2.1. Devaney had presented the whole overview of the metagenomics findings where astrovirus was a major co-infection virus in RSS. On further investigation the astrovirus existed as chicken astrovirus and avian nephritis virus infections. Our study’s purpose was to molecularly characterise ANV-3 genotype as an important co-factor in poorly performing/RSS chickens.
Lins 275 – 297: Do these similarities and differences with other available sequences make epidemiologic sense? And if they don’t, what is your explanation?
Lines 298 – 313: Do these hypothetical donor strains make epidemiological sense? And if they don’t, what is your explanation?
In answer to these two queries, the similarities, differences and parent donors may not appear to be epidemiological coherent. However, the putative recombination event parent donors, are not the specific progenitor strains. The donors speculated in this study are the most closely related to the recombinant donors within the sequences provided in the multiple sequence alignment. Further genomic screening and availability of circulating strains may help to add more epidemiological sense to the recombination signals found in this study.
Reviewer 2 Report
This work will provide the first molecular characterisation of the complete ANV-3 genome sequence isolated from a RSS-affected UK broiler flock. the results showed that a partial overlapping ORF1a and ORF1b and a non-overlapping ORF2, conserved domains and heptameric ribosomal frameshift (RFS) mechanism. This study can be used meaningful reference for others who is studying this virus.
the minor points: the quilty of the Fig 1,2,4,5,6 is not good enough, the letter can not been seen clearly.
the meaning of the number in Fig 5, 6 should be given
Author Response
REVIEWER 2
This work will provide the first molecular characterisation of the complete ANV-3 genome sequence isolated from a RSS-affected UK broiler flock. the results showed that a partial overlapping ORF1a and ORF1b and a non-overlapping ORF2, conserved domains and heptameric ribosomal frameshift (RFS) mechanism. This study can be used meaningful reference for others who is studying this virus.
the minor points: the quilty of the Fig 1,2,4,5,6 is not good enough, the letter can not been seen clearly.
the meaning of the number in Fig 5, 6 should be given
Dear reviewer,
I would like to thank you for taking the time to review this study and for providing the helpful comments.
The images have been all been resized and are uploaded into the document as PNG format with the required DPI. The fig 5 and fig 6 ANV number has been changed to ANV genotype number in the legend. Several of the astroviruses group nodes have been collapsed on the fig 6 tree allowing the tree format size to be expanded and is easier to see. Also the zip file of the images has been submitted to the journal as a PNG and PDF format.
Round 2
Reviewer 1 Report
I appears that the authors have responded to my comments and questions in separate document but have not addressed these in the manuscript.
Author Response
Dear Reviewer 1,
My reply has been uploaded as the attached word document and amendments to the manuscript have been highlighted in yellow (in a newly uploaded manuscript).
Yours sincerely,
Paula
